



**Chemical characteristics of PM$_{2.5}$: Impact of biomass burning at an**
**agricultural site of the North China Plain during a season of transition**
Linlin Liang[1], Guenter Engling[2,3], Chang Liu[1], Wanyun Xu[1], Xuyan Liu[4], Yuan Cheng[5], Zhenyu
Du[6], Gen Zhang[1], Junying Sun[1], Xiaoye Zhang[1]
[1] State Key Laboratory of Severe Weather & Key Laboratory for Atmospheric Chemistry, Chinese
Academy of Meteorological Sciences, Beijing 100081, China
[2] Division of Atmospheric Sciences, Desert Research Institute, Reno, NV 89512, USA
[3] Now at: California Air Resources Board, El Monte, CA 91731, USA
[4] National Satellite Meteorological Center, Beijing 100081, China
[5] School of Environment, Harbin Institute of Technology, Harbin 150001, China
[6] National Research Center for Environmental Analysis and Measurement, Beijing 100029 China
**Abstract:**
Biomass burning (BB) activities are ubiquitous in China, especially in North China, where
there is an enormous rural population and winter heating custom. In order to better understand
their impacts on aerosol chemical characteristics in rural and agricultural areas of the North China
Plain, BB tracers (i.e., levoglucosan (LG), mannosan (MN) and potassium (K$^+$)), as well as other
chemical components were quantified at a rural site (Gucheng, GC) from 15 October to 30
November, during a transition heating season, when the field burning of agricultural residues was
becoming intense. The measured daily average PM$_{2.5}$ concentrations of LG, MN and K$^+$ during
this study were 0.79 ± 0.75 µg m$^{-3}$, 0.03 ± 0.03 µg m$^{-3}$ and 1.52 ± 0.62 µg m$^{-3}$. Due to the
planetary boundary layer development, carbonaceous components and BB tracers showed higher
levels at nighttime than daytime, while OM and secondary inorganic ions were enhanced during
daytime, likely due to enhanced photochemical activity. An episode with high levels of BB tracers
was encountered at the end of October, 2016, with high LG at 4.37 µg m$^{-3}$. Based on the
comparison of chemical components during different BB periods, it appeared that biomass
combustion can obviously elevate carbonaceous components levels, whereas there seems to be
essentially no effect on secondary inorganic ions in the ambient air. Moreover, the LG/MN ratios
in different BB periods were consistent, while the LG/K$^+$ ratio during intensive BB periods was





significantly elevated at times, with $K^+$ not increasing as much as LG during intensive BB
episodes. This indicated that there were other sources of $K^+$ in the study region, such as fireworks,
fertilizer use, or soil resuspension, which don't have variable contributions of $K^+$ during the
intensive BB periods; however, local soft wood and vegetation combustion can't be excluded,
which have efficient formation of levoglucosan during flaming fires.
*Keywords*: Biomass burning; Organic tracers; Levoglucosan; Mannosan; Potassium

## 1. Introduction

Particulate air pollution is attracting more and more concerns in China because of their

obvious adverse impact on visibility reduction, as well as health implication and regional or global
climate change (Kanakidou et al., 2009; Pope and Dockery, 2006; Chen et al., 2017).
Carbonaceous species, i.e., organic carbon (OC) and elemental carbon (EC), and water-soluble
inorganic ions, e.g., $SO_4^{2-}$, $NO_3^-$ and $NH_4^+$ are the major components of ambient aerosols (Liang
et al., 2017; Du et al., 2014; Zheng et al., 2015; Tan et al., 2016). Biomass burning emissions
constitute a large source of ambient particulate pollution, especially for carbonaceous components,
i.e., primary organic carbon (POC) and black carbon (BC) on global scale (Bond et al., 2004; Tang
et al., 2018; Salma et al., 2017; Titos et al., 2017). As an important aerosol component, black
carbon from industrial and combustion emissions contributes to the enhanced $PM_{2.5}$ mass
concentrations and influences regional radiative forcing (Chen et al., 2017). Fresh biomass
burning aerosol was found to be mainly comprised of carbonaceous species which typically
constitutes 50-60% of the total particle mass (Hallquist et al., 2009). Yao et al. (2016) identified
approximately half of carbonaceous aerosols being contributed by biomass burning at Yucheng, a
rural site in the North China Plain.

Biomass burning emissions also represent a potentially large source of secondary organic

aerosol (SOA). The precursors and formation pathways of SOA from biomass burning emissions
were investigated by abundant field observations (e.g., Zhu et al., 2015; 2016; 2017; Adler et al.,
2011). Based on morphological particle analysis, Yao et al. (2016) investigated the smoke emitted
from biomass burning impacting SOA production. Sun et al. (2010) found that phenolic
compounds, which were emitted in large amounts from wood combustion, can form SOA at high



yields in aqueous-phase reactions. In addition, smoke from biomass burning can be transported
thousands of kilometers downwind from the source areas. Biomass burning aerosol from
Southeast Asia can be transported to China, Singapore and even further to North America (Liang
et al., 2017; Hertwig et al., 2015; Peltier et al., 2008). Based on molecular tracer measurements,
synoptic data as well as air mass back trajectory analysis, a fire episode was captured at a
background site of East China with smoke advected from Southeast Asia (Liang et al., 2017).

The North China Plain (NCP) is one of the most polluted regions in China. Severe haze–fog

of longer duration and more extensive coverage has occurred frequently in the NCP area,
especially during the seasons of autumn and winter. NCP covers one quarter of China's cultivated
land and yields 35% of the agricultural products in China (Boreddy et al., 2017). The rural
population in NCP is also large and dense, and biomass burning activities are common in this
region in form of cooking and heating. Intense fire activity typically occurs in October after the
corn harvest. Abundant smoke is emitted from agricultural burning, i.e., residential biofuel
combustion, open field burns, etc. Various field observations have investigated different aspects of
biomass burning, e.g., seasonal variations, chemical and physical properties of smoke particles,
spatial distribution, sources, transport, etc., in the NCP region (Cheng et al., 2013; Shen et al.,
2018; Sun et al., 2013; 2016; Boreddy et al., 2017; Yan et al., 2015). However, these field
investigations of the contribution of biomass burning to ambient aerosols in the NCP region were
concentrated on the city of Beijing (Cheng et al., 2013; Zheng et al., 2015; Duan et al., 2004).
Little field research about biomass burning was reported for rural areas in the NCP. In fact,
biomass burning activities are common in the rural areas of the NCP region, and the resulting
smoke aerosol can be transported to urban areas, e.g., the city of Beijing, resulting in haze
episodic events. Meanwhile, biomass burning studies at rural sites can provide valuable source
information of the biomass burning pollution in the North China region.

The objective of this study is to gain insights about the abundance of biomass burning smoke

during the autumn-winter transition season, following the corn harvest. In this paper, we focus on
quantifying multiple biomass burning tracers, i.e., LG, MN and $K^+$ as well as other chemical
species in $PM_{2.5}$ in GC during the autumn-winter transition biomass burning season. The study
results demonstrate the biomass burning pollution status in the rural atmosphere of North China



and explore the impact of biomass burning activities on the chemical properties of ambient
aerosols.

## 2. Site description and experimental Methods

### 2.1 Site description

Samples were collected at a rural site, Gucheng (GC, 39°09'N, 115°44'E; 15.2 m a.s.l),

located on a platform at the China Meteorological Administration farm in the town of Gucheng
(GC site), approximately 110 km southwest of Beijing and 35 km north of the city of Baoding
(population of about 5 million) in Hebei province, as shown in *Fig. S1*. The station is surrounded
by agricultural fields, with major crop species being corn and wheat. The dominant wind direction
at GC is southwest and northeast during the study period. This site is upwind of Beijing, when the
wind blows from the south or southwest, where heavily polluted cities and regions of Hebei
province, i.e., Baoding, Shijiazhuang, Xingtai, Handan, are located. Thus, it is an appropriate
station for representing the air pollution situation in the NCP region (Sheng et al., 2018; Chi et al.,
2018; Xu et al., 2019; Kuang et al., 2020).

Daytime and nighttime $PM_{2.5}$ samples were collected from 15 October, 2016 to 23 November,

2016, by using $PM_{2.5}$ High-volume (Hi-Vol) samplers (GUV-15HBL1, Thermo Fisher Scientific
CO., LTD), at the nominal flow rate of 1.13 $m^3$ $min^{-1}$. All $PM_{2.5}$ samples were collected on quartz
fiber filters, prebaked at 850 °C for at least 5 h to remove organic material. A total of 33 couples of
daytime/nighttime samples and 6 whole-day samples as well as 4 field blank samples were
collected during the sampling period. The filters were stored at -20 °C after sample collection.

### 2.2 Experimental Methods

#### 2.2.1 Anhydrosugar and water-soluble inorganic ion analysis

The quartz filter samples were analyzed for biomass burning anhydrosugar tracers, i.e., LG

and MN, using an improved high-performance anion-exchange chromatography (HPAEC) method
with pulsed amperometric detection (PAD) on a Dionex ICS-5000+ system. LG and MN were
separated by a Dionex Carbopac MA1 analytical column and guard column with an aqueous
sodium hydroxide (NaOH, 480 mM) eluent at a flow rate of 0.4 mL $min^{-1}$. The detection limit of
LG and MN was 0.002 mg $L^{-1}$ and 0.005 mg $L^{-1}$, respectively. More details about the



HPAEC-PAD method can be found elsewhere (Iinuma et al., 2009).

The quartz filter samples were also analyzed for water-soluble inorganic ions by a Dionex

ICS-5000+ ion chromatograph, including three anions (i.e., $SO_4^{2-}$, $NO_3^-$, $Cl^-$,) and five cations
(i.e., $NH_4^+$, $Ca^{2+}$, $Na^+$, $K^+$ and $Mg^{2+}$). The cations were separated on an Ionpac CS12 analytical
column and CG12 guard column with a 20 mM methanesulfouic acid as eluent at a flow rate of
1.0 mL min$^{-1}$, while the anions were separated on an Ionpac AS11-HC column and AG11-HC
guard column with 21.5 mM KOH eluent at a flow rate of 1.0 mL min$^{-1}$. The water-soluble
inorganic ion data were corrected by field blanks.
**2.2.2 Organic carbon/elemental carbon analysis**

OC and EC were measured on a punch (0.526 cm$^2$) of each quartz sample by a

thermal/optical carbon analyzer (DRI Model 2001, Desert Research Institute, USA), using the
Interagency Monitoring of Protected Visual Environments (IMPROVE) thermal evolution
protocol with reflectance charring correction. The analytical error of OC was within 10%, and one
sample of every 10 samples was selected at random for duplicate analysis. The detection limit of
OC was 0.82 µg cm$^{-2}$ (Liang et al., 2017).
**2.2.3 Gas online monitoring (i.e., NO, NO$_2$, SO$_2$, O$_3$, CO and NH$_3$)**

During this campaign, commercial instruments from Thermo Fisher Scientific Co., LTD were

used to measure O$_3$ (TE 49C), NO/NO$_2$/NOx (Model 42CTL), CO (TE 48CTL), and SO$_2$
(TE43CTL), while NH$_3$ was measured by an ammonia analyzer (DLT-100, Los Gatos Research,
USA) at GC station. All measurement data quality was controlled according to standards described
elsewhere (Xu et al., 2019; Lin et al., 2011; Meng et al., 2018; Ge et al., 2018).
**2.2.4 Meteorological parameters**

The meteorological parameters, including air temperature, relative humidity (RH) and wind

speed on a 24-h resolution at the GC site are presented in Fig. 1. The wind speed was usually
lower than 1 m s$^{-1}$ at the GC site, indicating that calm wind was most frequent, and unfavorable
dispersion conditions prevailed during the autumn-winter transition season. During the sampling
time from 15 October, 2016 to 23 November, 2016, the mean RH at GC was observed at 77%,
exhibiting moist conditions. These meteorological parameters indicate that GC was characterized
by humid and stagnant air masses.





**2.2.5 Back trajectory and fire spot analysis**
To characterize the transport pathways of the aerosol at the Gucheng site, back-trajectories
were calculated with the NOAA Hybrid Single-Particle Lagrangian Integrated Trajectory
(HYSPLIT) model via NOAA ARL READY Website (http://ready.arl.noaa.gov/HYSPLIT.php).
To investigate the influence of biomass burning activities in surrounding areas, fire hot spot
counts were obtained from the Fire Information for Resource Management System (FIRMS)
(available at https://firms.modaps.eosdis.nasa.gov/download/).
**3. Summary and implications**
**3.1 Characteristics of chemical components in $PM_{2.5}$**
Fig. 1 describes the time-series variation obtained for daily $PM_{2.5}$, and its major
components (OC, EC, $SO_4^{2-}$, $NO_3^-$ and $NH_4^+$), biomass burning tracers (LG, MN and $K^+$) and
meteorological factors (temperature, RH, wind speed and rainfall) during the sampling period. The
mean concentrations and standard deviation of $PM_{2.5}$, the quantified components and
meteorological parameters for the whole study period are listed in Table 1. In this study, the mass
concentration of $PM_{2.5}$ was reconstituted by the sum of carbonaceous components (1.6×OC + EC)
and inorganic ions ($SO_4^{2-}$ + $NH_4^+$ + $NO_3^-$ + $Cl^-$ + $Ca^{2+}$ + $Na^+$ + $K^+$ + $Mg^{2+}$). The average daily
$PM_{2.5}$ mass concentration in the autumn-winter transition season at GC reached $137 \pm 72.4$ µg m$^{-3}$,
ranging from 23.3 µg m$^{-3}$ to 319 µg m$^{-3}$, which is higher than during the severe winter haze in
January, 2013 at an urban site in Beijing (121.0 µg m$^{-3}$) (Zheng et al., 2015). The mass
concentrations of these chemical species during the day are distributed as follows (from highest to
lowest): OC > EC > $NO_3^-$ > $SO_4^{2-}$ > $NH_4^+$ > $Cl^-$ > $Ca^{2+}$ > $K^+$ > $Na^+$ > $Mg^{2+}$. Organic matter (OM)
was the most abundant component, the daily average value of which was $70.4 \pm 49.6$ µg m$^{-3}$,
accounting for nearly half (46.7%) of $PM_{2.5}$ mass, indicating obvious organic pollution at the rural
site in the North China Plain during the sampling season.
Secondary inorganic aerosol (sulfate, $SO_4^{2-}$; nitrate, $NO_3^-$ and ammonium, $NH_4^+$, SNA)
species, were the major water soluble ions, accounting for 82.8% of total water soluble ions, the
daily average values of which were $10.5 \pm 6.87$ µg m$^{-3}$, $15.9 \pm 9.29$ µg m$^{-3}$ and $10.9 \pm 5.51$ µg m$^{-3}$
(Table 1). SNA species exhibited a synchronous temporal trend, while the $NO_3^-$ concentrations



exceeded those of $SO_4^{2-}$ at the GC site, in contrast to the results of previous studies, e.g., Tan et al.
(2016), who found $SO_4^{2-}$ to be the dominant species in $PM_{2.5}$ during winter time in 2006 in Beijing.
$SO_4^{2-}$ has previously been reported for many sites to be the dominant component of SNA in $PM_{2.5}$,
followed by $NO_3^-$ and $NH_4^+$ (Duan et al., 2003; Yang et al., 2011; He et al., 2012). These findings
can likely be explained by the variation in $SO_2$ and NOx emissions over the last decade in China
(Sun et al., 2016a; 2016b). $SO_2$ emissions have decreased as a result of desulfurization, whereas
NOx emissions have increased as a result of industrialization and an increase in the number of
vehicles. Similarly, Chi et al., (2018) also found $NO_3^-$ concentrations exceeded those of $SO_4^{2-}$ at
both Beijing and GC sites during the winter time in 2016, although they observed that $NH_4^+$ was
the dominant component of SNA (the concentrations of $SO_4^{2-}$, $NO_3^-$ and $NH_4^+$ were 14.0 μg m$^{-3}$,
14.2 μg m$^{-3}$, and 24.2 μg m$^{-3}$, respectively).

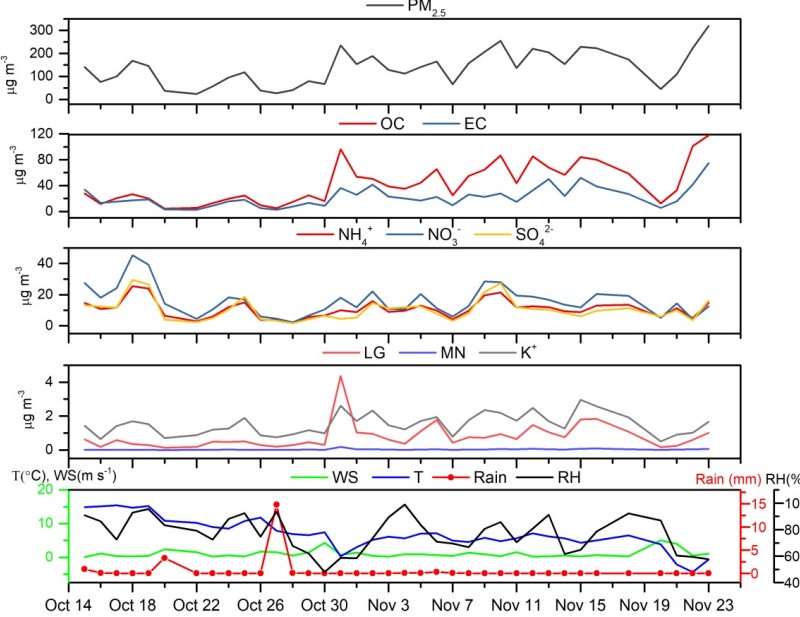


Fig. 1. Time-series variation obtained for $PM_{2.5}$ and its major components (OC, EC, $SO_4^{2-}$, $NO_3^-$ and $NH_4^+$),
biomass burning tracers (LG, MN and K$^+$) and meteorological factors (temperature, RH, wind speed and rainfall)
at the GC site during the sampling period from 15 Oct to 23 Nov 2016.
The measured daily average concentrations of biomass burning tracers, i.e., LG, MN and K$^+$ in
$PM_{2.5}$ during our study were 0.79 ± 0.75 μg m$^{-3}$, 0.03 ± 0.03 μg m$^{-3}$ and 1.52 ± 0.62 μg m$^{-3}$ (Table
1). The anhydrosugar levels (LG and MN) in this study were both higher than those observed in


the city of Beijing during summer and winter seasons (Cheng et al., 2013; Yan et al., 2015). The
highest concentrations of LG in Gucheng were observed on 31 October, 2016 with 4.37 μg m$^{-3}$,
which is a sharp increase (over 30 times) of the minimum concentration (0.14 μg m$^{-3}$) during that
period. Accordingly, the PM$_{2.5}$ concentration during that period was also elevated (as high as 236
μg m$^{-3}$) (Fig. 1).
Table 1.   Average concentrations and the range of chemical components in PM$_{2.5}$ (μg m$^{-3}$) and meteorological
data observed at GC site from 15/10/2016 to 23/11/2016.

| Species | Daytime (N = 34) | | Nighttime (N = 33) | | Whole period (N = 37) | |
|---|---|---|---|---|---|---|
| | Concentration | Range | Concentration | Range | Concentration | Range |
| PM$_{2.5}$ mass | 117 ± 58.8 | 19.0 - 225 | 170 ± 116 | 21.1 - 465 | 137 ± 72.4 | 23.3 - 319 |
| OC | 26.8 ± 15.7 | 3.78 - 64.8 | 61.6 ± 49.5 | 2.88 - 175 | 44.0 ± 31.0 | 4.13 - 117 |
| EC | 13.4 ± 8.49 | 1.44 - 34.0 | 30.9 ± 28.5 | 2.21 - 129 | 21.7 ± 15.8 | 2.46 - 74.9 |
| TC | 49.3 ± 27.6 | 5.76 - 124 | 92.5 ± 73.6 | 5.10 - 289 | 65.8 ± 44.1 | 7.36 - 192 |
| OC/EC | 2.02 ± 1.26 | 1.09 - 3.31 | 2.25 ± 1.04 | 1.04 - 6.72 | 1.95 ± 0.60 | 0.83 - 3.10 |
| SO$_4^{2-}$ | 12.1 ± 9.31 | 1.65 - 39.7 | 9.02 ± 6.22 | 1.55 - 23.2 | 10.5 ± 6.87 | 1.66 - 29.5 |
| NO$_3^-$ | 16.9 ± 9.96 | 1.85 - 41.2 | 13.1 ± 8.52 | 1.56 - 38.0 | 15.9 ± 9.29 | 2.40 - 45.2 |
| Cl$^-$ | 4.33 ± 2.30 | 0.82 - 9.46 | 6.08 ± 4.00 | 0.62 – 16.0 | 4.90 ± 2.46 | 0.93 - 9.37 |
| NH$_4^+$ | 11.7 ± 6.76 | 1.84 - 26.0 | 10.0 ± 5.75 | 1.33 - 22.2 | 10.9 ± 5.51 | 1.99 - 25.4 |
| K$^+$ | 1.43 ± 0.54 | 0.20 - 2.64 | 1.78 ± 0.95 | 0.22 - 4.19 | 1.52 ± 0.62 | 0.50 - 2.96 |
| Mg$^{2+}$ | 0.26 ± 0.14 | 0.07-0.64 | 0.19 ± 0.09 | 0.06 - 0.38 | 0.14 ± 0.12 | 0.04 - 0.43 |
| Ca$^{2+}$ | 2.24 ± 1.01 | 1.02-4.75 | 1.56 ± 0.08 | 0.77 - 3.56 | 1.54 ± 0.90 | 0.49 - 3.84 |
| Na$^+$ | 0.44 ± 0.17 | 0.10 - 0.79 | 0.43 ± 0.24 | 0.10 - 1.31 | 0.42 ± 0.17 | 0.11 - 0.88 |
| NO$_3^-$/ SO$_4^{2-}$ | 1.67 ± 0.82 | 0.75 - 5.52 | 1.54 ± 0.57 | 0.74 - 3.50 | 1.65 ± 0.62 | 0.78 - 3.96 |
| Levoglucosan | 0.57 ± 0.62 | 0.05 - 3.74 | 1.10 ± 0.99 | 0.05 - 4.82 | 0.79 ± 0.75 | 0.14 - 4.37 |
| Mannosan | 0.024 ± 0.023 | 0.00 - 0.14 | 0.05 ± 0.04 | 0.00 - 0.21 | 0.03 ± 0.03 | 0.00 - 0.18 |
| NO (ppb) | 23.0 ± 14.7 | 2.07 - 56.0 | 45.9 ± 29.5 | 1.59 - 96.9 | 31.8 ± 18.3 | 1.81 - 68.5 |
| NO$_2$ (ppb) | 25.8 ± 10.4 | 8.18 - 51.6 | 29.3 ± 9.37 | 8.81 - 51.1 | 26.6 ± 8.74 | 8.62 - 51.4 |
| SO$_2$ (ppb) | 9.78 ± 4.96 | 3.11 - 22.5 | 9.63 ± 5.67 | 2.91 - 28.7 | 8.61 ± 4.04 | 3.37 - 20.4 |
| CO (ppm) | 0.96 ± 0.73 | 0.03 - 2.49 | 1.29 ± 1.04 | 0.02 - 3.26 | 1.05 ± 0.76 | 0.12 - 2.48 |
| O$_3$ (ppb) | 13.0 ± 9.10 | 1.42 - 41.84 | 5.00 ± 5.73 | 1.60 - 24.30 | 9.25 ± 5.78 | 1.67 - 24.0 |
| NH$_3$ (ppb) | 16.4 ± 11.3 | 1.68 - 46.2 | 18.3 ± 10.7 | 1.03 - 42.7 | 17.1 ± 9.88 | 1.46 - 44.4 |
| Temperature (℃) | 7.71 ± 4.01 | -2.07-15.9 | 3.30 ± 4.69 | -6.60 - 14.5 | 6.95 ± 4.58 | -4.33 - 15.4 |
| Relative Humidity (%) | 68 ± 17 | 31 - 98 | 85 ± 14 | 34 - 100 | 77 ± 13 | 48 - 99 |
| Wind speed (m s$^{-1}$) | 1.43 ± 1.17 | 0.09 - 5.65 | 0.79 ± 1.55 | 0.03 - 7.19 | 1.07 ± 1.14 | 0.04 - 5.02 |

During this campaign, the daily average RH value was observed at 77 ± 13%, with a range
from 48% to 99%, while the daily average wind speed averaged at 1.07 ± 1.14 m s$^{-1}$, exhibiting
moist and stable synoptic conditions at this rural site. Under low wind speeds and high RH,
aerosol particles are very conducive to accumulate in the atmosphere. Moreover, there was rare
precipitation during the sampling period at the GC site, except for two days, i.e., 20 and 27


October, 2016. High wind speed and precipitation can interrupt stable synoptic meteorological
conditions, enhancing the dilution and dispersion of pollutants in the air, and ultimately increased
wind speed and rainfall will cause opposite $PM_{2.5}$ patterns (Fig. 1).
**3.2 Day-night variations in the characteristics of $PM_{2.5}$ chemical components**

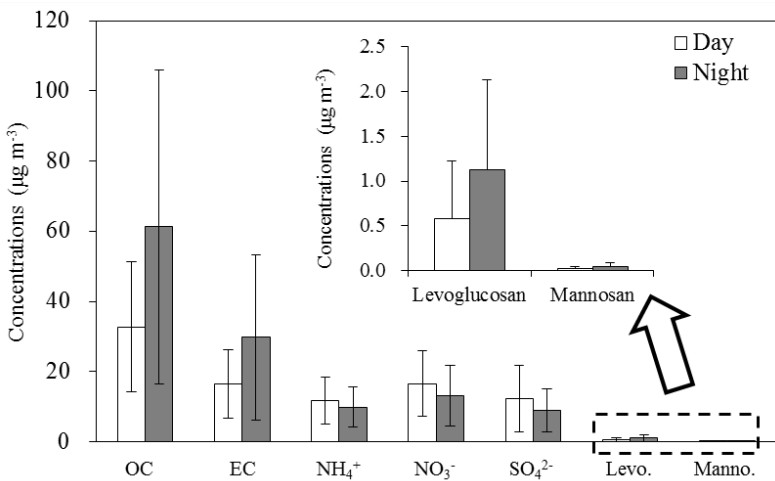


Fig. 2. Day and night distributions of mean concentrations of main chemical components in $PM_{2.5}$ observed at GC
site during the sampling period.
The corresponding average mass concentrations and percent contributions of individual
chemical components to total estimated $PM_{2.5}$ mass in daytime and nighttime during this
campaign are reported in Fig. 2 and Fig. 3. Time-series variations of $PM_{2.5}$ and individual
components in daytime and nighttime during the sampling period are shown in *Fig. S2*. Generally,
carbonaceous components and biomass burning tracers exhibited higher levels during nighttime
than daytime, while secondary inorganic ions showed the opposite pattern, i.e., higher
concentrations during daytime than nighttime. In addition, the gap of carbonaceous components
and anhydrosugars between daytime and nighttime (two-fold) was more significant than for
secondary inorganic ions. That may be due to the variations of pollutant concentrations not only
being controlled by the chemical reactions but also being subject to the influence of the planetary
boundary layer (PBL) development. In the night, the PBL height decreases, compressing air
pollutants into a shallow layer, and subsequently resulting in faster accumulation and higher
concentrations of pollutants (Zheng et al., 2015; Zhong et al., 2018; 2019). Carbonaceous





components and anhydrosugars are not subject to significant differences in chemical reactions in
the ambient air between daytime and nighttime; thus, they will be mainly influenced by the
variations of the PBL height. Correspondingly, the contributions of OM and EC to $PM_{2.5}$ were
observed to be higher at nighttime (53.9% and 16.6%) than daytime (43.8% and 13.7%). The
contribution of LG to $PM_{2.5}$ during nighttime (0.64%) was also higher than that during daytime
(0.37%). However, unlike OM, secondary inorganic ions have an important formation path, i.e.,
photochemical processing, during daytime. Thus, the secondary species ($SO_4^{2-}$, $NO_3^-$ and $NH_4^+$)
were enhanced during daytime due to photochemical formation (Fig. 2 and 3). Moreover, such an
enhancement in secondary transformations at daytime is more evident in terms of the mass
contributions of secondary inorganic ions to $PM_{2.5}$, that the contributions of $SO_4^{2-}$, $NO_3^-$ and $NH_4^+$
to $PM_{2.5}$ decreased from daytime (9.9%, 14.5% and 10.0%) to nighttime (6.5%, 9.6% and 7.1%).

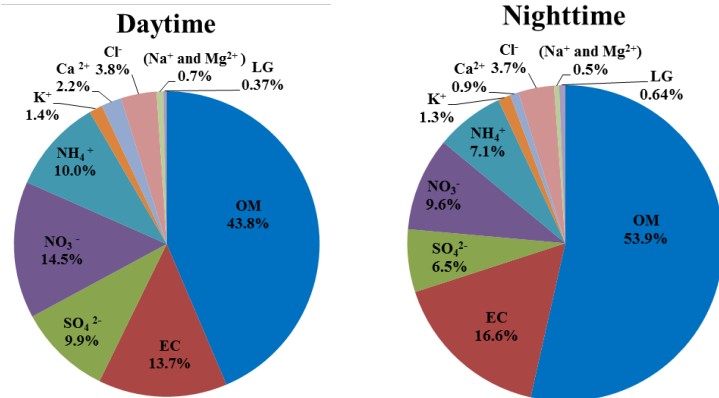


Fig. 3. Percent contributions of individual component mass concentrations to total estimated $PM_{2.5}$ mass in
daytime and nighttime during the sampling period.
In addition, the concentrations of other water-soluble inorganic ions, i.e., $K^+$ and $Cl^-$ during
nighttime (1.78 ± 0.95 μg m$^{-3}$ and 6.08 ± 4.00 μg m$^{-3}$) were higher than those in daytime (1.43 ±
0.54 μg m$^{-3}$ and 4.33 ± 2.30 μg m$^{-3}$), while their contributions to $PM_{2.5}$ were reversed, due to the
significant accumulation and higher concentrations of pollutants during nighttime. The
contribution of the primary sources of $Ca^{2+}$, $Mg^{2+}$ and $Na^+$ in nighttime were lower than those
during daytime, especially for $Ca^{2+}$, decreasing from 2.2% in daytime to 0.9% at nighttime (Fig. 3).
This may be because of these inorganic ions being emitted from primary sources, such as dust, soil
resuspension and sea salt, which are subject to more activity during the daytime and also
influenced by the airflow dynamics.





### 3.3 Biomass burning episodes and the impacts on chemical PM$_{2.5}$ characteristics

Table 2. Concentrations of chemical components in PM$_{2.5}$ aerosols and gaseous species collected at the GC site during the three biomass burning periods from 15 Oct to 23 Nov 2016.

| Species | Period I (15-30 Oct) Minor biomass burning | Period II (31 Oct) Intensive biomass burning | | Period III (1 -23, Nov) Major biomass burning | |
| --- | --- | --- | --- | --- | --- |
| | Concentration | Concentration | Ratio* | Concentration | Ratio* |
| PM$_{2.5}$ | 81.0 ± 44.5 | 235 | 2.91 | 172 ± 62.4 | 2.12 |
| Levoglucosan | 0.36 ± 0.14 | 4.37 | 12.1 | 0.92 ± 0.47 | 2.56 |
| Mannosan | 0.015 ± 0.005 | 0.18 | 12.0 | 0.042 ± 0.02 | 2.80 |
| OC | 16.2 ± 7.52 | 96.3 | 5.93 | 59.9 ± 25.3 | 3.69 |
| EC | 12.2 ± 5.85 | 36.0 | 2.96 | 29.1 ± 15.8 | 2.39 |
| TC | 28.4 ± 13.1 | 132 | 4.66 | 89.2 ± 38.8 | 3.14 |
| SO$_4^{2-}$ | 10.3 ± 8.96 | 4.56 | 0.44 | 10.9 ± 5.55 | 1.06 |
| NO$_3^-$ | 16.6 ± 12.9 | 18.1 | 1.09 | 15.2 ± 6.48 | 0.92 |
| NH$_4^+$ | 10.1 ± 7.40 | 10.0 | 0.99 | 11.4 ± 4.19 | 1.13 |
| K$^+$ | 1.16 ± 0.36 | 2.61 | 2.25 | 1.72 ± 0.62 | 1.48 |
| Cl$^-$ | 3.46 ± 1.97 | 7.49 | 2.16 | 5.81 ± 2.33 | 1.68 |
| OC/EC | 1.53 ± 0.35 | 2.67 | 1.75 | 2.22 ± 0.53 | 1.45 |
| NO$_3^-$/ SO$_4^{2-}$ | 1.74 ± 0.60 | 3.96 | 2.28 | 1.47 ± 0.39 | 0.84 |
| LG/OC | 0.025 ± 0.008 | 0.045 | 1.80 | 0.015 ± 0.006 | 0.60 |
| LG/EC | 0.039 ± 0.019 | 0.121 | 3.10 | 0.041 ± 0.027 | 1.05 |
| LG/MN | 24.9 ± 4.44 | 24.1 | 0.97 | 22.7 ± 6.71 | 0.91 |
| LG/K$^+$ | 0.36 ± 0.081 | 1.67 | 4.64 | 0.52 ± 0.76 | 1.44 |
| NO (ppb) | 21.7 ± 12.5 | 21.7 | 1.00 | 39.5 ± 18.6 | 1.82 |
| NO$_2$ (ppb) | 21.8 ± 4.95 | 26.5 | 1.22 | 30.0 ± 9.18 | 1.38 |
| NO$_X$ (ppb) | 43.6 ± 16.3 | 48.2 | 1.11 | 69.5 ± 24.5 | 1.59 |
| SO$_2$ (ppb) | 5.83 ± 2.46 | 8.04 | 1.38 | 10.6 ± 3.90 | 1.82 |
| CO (ppm) | 0.44 ± 0.33 | 0.70 | 1.59 | 1.51 ± 0.67 | 3.43 |
| O$_3$ (ppb) | 9.79 ± 4.88 | 23.2 | 2.37 | 8.21 ± 5.47 | 0.84 |
| NH$_3$ (ppb) | 14.3 ± 6.12 | 11.1 | 0.78 | 19.5 ± 10.8 | 1.36 |

*: indicates that the ratios of the intense BB period or major biomass burning period were divided by those from the minor BB period.

An episode with high biomass burning tracer levels was encountered on 31 October, 2016. The concentrations of levoglucosan in PM$_{2.5}$ during this one-day episode (4.37 µg m$^{-3}$) were significantly higher than those during typical transition season at the GC site (0.69 ± 0.47 µg m$^{-3}$) (Fig.1). Here, we mainly distinguish three sub-periods based on daily LG concentrations during the time frame from 15 October to 23 November, 2016. The three periods were separated as follows: 15-30 October (Period I: Minor biomass burning), 31 October (Period II: Intensive biomass burning), 1- 23 November (Period III: Major biomass burning). Table 2 compares the concentrations of PM$_{2.5}$ mass, chemical components and gases at the GC site during these three



periods, as well as the ratios between the intensive and major BB periods to minor BB period.
Compared to typical autumn-winter transition time, the level of LG during the intensive BB
episode was about 12 times of that during the minor BB period. Furthermore, the concentrations
of OC and EC were also increased in the intensive and major BB periods (Table 2). For example,
during the intensive BB episode, OC was nearly 6 times of that during the minor BB period,
demonstrating that biomass burning influence can obviously contribute to carbonaceous aerosols
in the ambient rural environment. The episode on 31 October, 2016 with high $PM_{2.5}$ levels was
apparently caused by intensive biomass combustion activities in the North China Plain.

During Period I, LG and MN were at low levels, with average concentrations of $0.36 \pm 0.14$

and $0.015 \pm 0.005$ µg m$^{-3}$. When entering into November, the heating season in the North China
region was commencing , resulting in the ambient levels of LG and MN increase to $0.92 \pm 0.47$
and $0.042 \pm 0.02$ µg m$^{-3}$ during period III, about 3 times of those in Periods I. Due to the frequent
heating activities in form of straw burning, we found the concentrations of $PM_{2.5}$ mass,
carbonaceous components, $K^+$ and $Cl^-$ strongly increased during period III. Ambient
concentrations of OC and EC, for example, increased from $16.2 \pm 7.52$ µg m$^{-3}$ and $12.2 \pm 5.85$ µg
m$^{-3}$ on average during the minor biomass burning period I to $59.9 \pm 25.3$ µg m$^{-3}$ and $29.1 \pm 15.8$
µg m$^{-3}$ in the major biomass burning period III.

However, compared to the carbonaceous components, the secondary inorganic aerosol

species ($SO_4^{2-}$, $NO_3^-$, $NH_4^+$) exhibited a different pattern, i.e., showing no obvious differences
between BB period I and periods II and III. The ratios of $SO_4^{2-}$, $NO_3^-$, $NH_4^+$ during periods II and
III to period I were all around 1.0 (Tab. 2), with no increasing trend. Moreover, the relationships
between LG and OC, EC during daytime and nighttime were both better than those with SNA (*Fig.*
*S3).* This pattern implied that biomass burning can evidently elevate the levels of carbonaceous
components but have no significant effect on secondary organic ions in the ambient air. This
finding is similar to the observations at Mt. Tai in China, where the concentrations of $NH_4^+$ and
$SO_4^{2-}$ were both higher in the minor BB period than those in the major BB period, while the
concentration pattern of $NO_3^-$ was reversed, i.e., higher in the major BB period compared to the
minor BB period (Boreddyet al., 2017). However, the precursor gases of SNA, i.e., $SO_2$, NO, $NO_2$
and $NH_3$, were observed to have an increasing trend when biomass burning was prevalent during



periods II and III (the ratios of precursor gases of SNA during periods III to period I were in the
range of 1.38 to 1.82) (Table 2). The time-series variations of the gases ($SO_2$, NOx, $NH_3$, CO and
$O_3$) and PBL during the sampling period are shown in ***Fig. S4.*** The primary emission gases were
exhibited negative relationships with PBL, while $O_3$ exhibited obvious positive relationship with
PBL. The average concentration of CO clearly increased from $0.44 \pm 0.33$ ppm in period I to 1.51
$\pm 0.67$ ppm in period III, which illustrates that biomass burning was an important resource for CO
in the ambient air (Tab. 2), similar to the findings of Jung et al. (2014) observed at Daejeon, Korea,
during the rice harvest period.

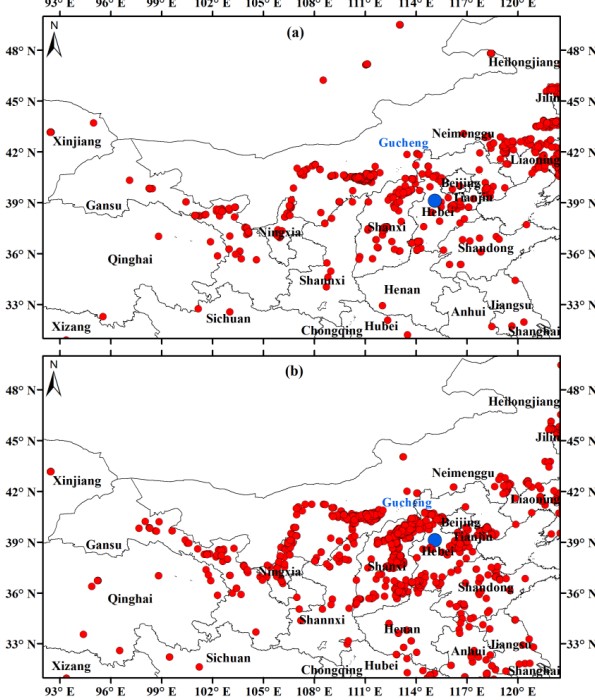


Fig.4. Fire spots at GC site and the surrounding provinces from (a) 15-30 October, 2016 and (b) 1 -23, November,
2016, observed by MODIS Terra satellites (blue dot is GC station).

The combustion of biomass, especially of agricultural residues (e.g., wheat and corn straw)

is very common in the rural areas in North China during the autumn-winter transition period.
During the autumn harvest season in North China, wheat and corn straw burning is common
practice, resulting in more abundant fire spots observed during period III than period I (Fig. 4).
The intense biomass burning event on 31 October, 2016 was also supported by air mass back



trajectory analysis (Fig. 5), performed with the TrajStat software. Based on the 48 h back
trajectories at the GC site at 00:00 (UTC time) on 1 November, 2016, the air mass at the GC site
was restricted in the region of Bejing-Tianjing-Hebei, the polluted area where fire spots were
numerous. However, on the previous and following day of this episode, i.e., 31 October and 2
November, 2016 onward, the air masses arriving in GC were advected from Northwest Mongolia,
where mostly desert areas are present, with less farm land and rare biomass burning activities (Fig.

5).

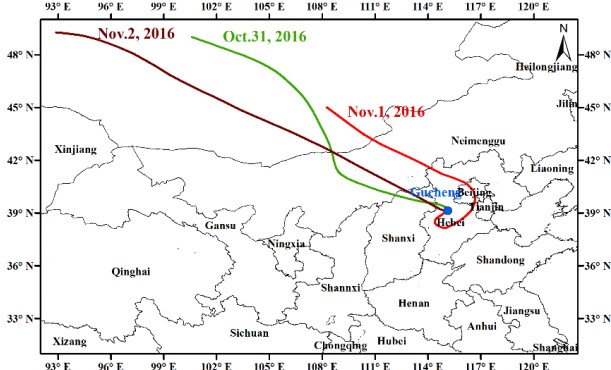


Fig. 5. 48 h back trajectories at 500 m at GC site from 31 October to 2 November, 2016.

Mean percentiles of major components in $PM_{2.5}$ with respect to different biomass burning

pollution periods at GC site during the sampling time are shown in Fig. 6. With the variation of
biomass burning pollution periods, the EC fraction seems to exhibit no obvious change and the
OC fraction increased significantly, while the contributions of sulfate, nitrate and ammonium to
$PM_{2.5}$ all decreased sharply (Fig. 6). This suggests that organic aerosol species become more
important for biomass burning pollution periods, concerning their contribution to the $PM_{2.5,}$ while
EC has no such character. The OM percentage during intense biomass burning period II was
65.4%, about double of that during the minor biomass burning period (34.0%), indicating that
there was a large fraction of OM in $PM_{2.5}$ originating from biomass burning at the GC site during
intensive BB period II. Opposite to OM, contributions of secondary inorganic ions to $PM_{2.5}$
significantly decreased with the biomass burning pollution becoming more severe. The
contributions of $SO_4^{2-}$, $NO_3^-$ and $NH_4^+$ to $PM_{2.5}$ during the minor BB episode (11.6%, 20.5% and
12.5%) obviously declined during the intense BB episode (1.73%, 7.73% and 4.24%). This



phenomenon further illustrates that biomass burning emissions can substantially increase the
ambient levels of organic aerosol, while not affecting the contributions of secondary inorganic
aerosols.

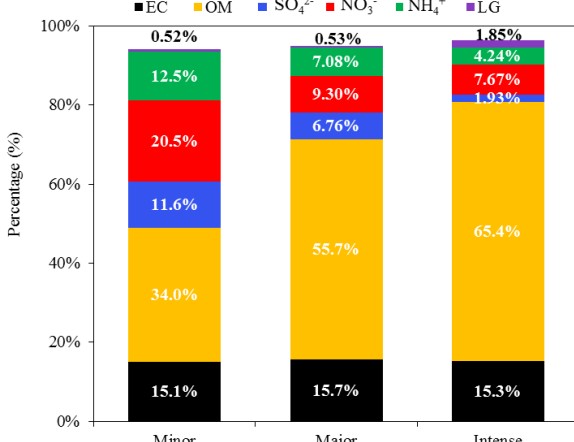

Fig. 6. Mean percentiles of major components in PM$_{2.5}$ with respect to different biomass burning pollution periods

at GC site during the sampling time.

**3.4 Relationships among tracers during different biomass burning pollution**
**periods**

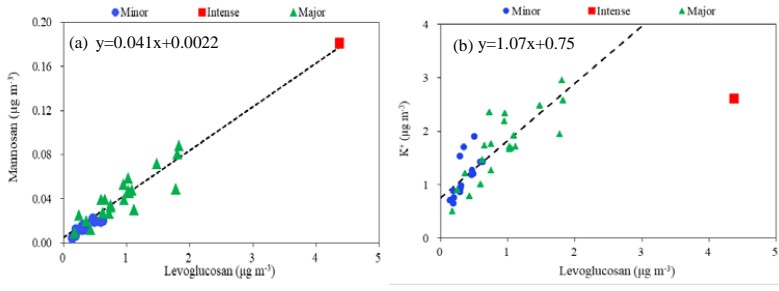

Fig. 7. Scatter plots of (a) levoglucosan versus mannosan, (b) levoglucosan versus K$^+$

The LG/MN ratios during minor, major and intense biomass pollution periods were
observed at high values, i.e., 24.9, 24.1, 22.6, respectively (Tab. 2). However, the LG/K$^+$ ratios
during the three periods (minor, major and intense biomass burning periods) varied considerably
(0.36, 0.52, 1.67) (Tab. 2). The scatter plots between LG and MN and K$^+$ are shown in Fig. 7. The
relationship between LG and MN in the three periods followed one line, especially for period II





(red dot), which was located far from the concentrated plots of periods I and III, yet it was still on
the same regression line of the fitting curve (Fig. 7a). In contrast, the relationship between LG and
$K^+$ did not show a consistent pattern as that of LG versus MN, with the red dot of period II being
off from the fitted regression line (Fig. 7b). The LG/$K^+$ ratios during minor and major BB periods
(0.36 and 0.52) were similar to those during a biomass burning episode at an urban site of Beijing
during summer time (Lev/$K^+$ = 0.51), an urban site in Guangzhou (0.29) and an suburban site in
Zhuhai (0.4) during the dry season (Zhang et al., 2015). This LG/$K^+$ ratio during the intense BB
period II was observed at 1.67, which was significantly higher than that in periods I and III.
This high value of LG/$K^+$ ratio is probably the most representative of the local BB aerosol,
similar to the smoke aerosols from soft wood combustion (China fir and red pine) (Sang et al.,
2013; 2020). Similarly, the LG/$K^+$ ratios observed in Austria were also in the range of 0.91 to 1.7
during winter time, which were attributed mainly to wood burning in households (Caseiro et al.,
2009). Based on the results of biomass source combustion studies (Engling et al., 2009; Chantara
et al., 2019), comparing LG to $K^+$, it appears that there is a large enrichment of LG in wood or
vegetation burning with efficient formation of LG during the flaming phase. There was a time of
strong process decrease in temperature at Gucheng site, the average daily temperature was sharply
decreased from 7.5°C at 30 Oct, 2016 to 0.31°C at 31 Oct, 2016 and the average temperature at the
night of 31 Oct, 2016 was even decreased to -3.4°C (Fig.1). Thus, it must be many combustion
taken places around the sampling site for heating, by burning straws, branches, as well as local
soft woods, since these fuels are also commonly used in rural areas of North China, i.e., pine,
poplar, China fir, etc. Moreover, the reason for this phenomenon maybe also due to other sources
of $K^+$ in the study region, such as fireworks, fertilizers, soil resuspension, etc. (Drewnick et al.,
2006; Urban et al., 2012; Cheng et al., 2013). During intensive biomass combustion periods, the
contributions from other sources of $K^+$ exhibit no or little variation, causing $K^+$ to not increase as
rapidly as LG, which is a unique source tracer from biomass burning in ambient aerosols.
**4. Summary and implications**
Biomass burning activities are ubiquitous in China, especially in North China, where
enormous rural populations are present, with a common winter heating custom. Anhydrosugars,
including levoglucosan and mannosan, and water-soluble potassium ion were employed as



molecular tracers to investigate the characteristics of biomass burning activities as well as their
impact on chemical properties of ambient aerosols in rural areas of North China. The measured
daily average concentrations of LG, MN and $K^+$ in $PM_{2.5}$ during a transition heating season, from
15 October to 30 November, 2016 were $0.79 \pm 0.75$ µg m$^{-3}$, $0.03 \pm 0.03$ µg m$^{-3}$ and $1.52 \pm 0.62$ µg
m$^{-3}$. The daily PBL development caused carbonaceous components and biomass burning tracers to
be higher at nighttime than daytime, while the patterns of secondary inorganic ions ($SO_4^{2-}$, $NO_3^-$
and $NH_4^+$) were opposite, which were enhanced during daytime due to photochemical formation.
Due to intense emissions and contribution of stable synoptic conditions, an episode with extreme
biomass burning tracer levels was encountered on 31 October, 2016, with concentrations of LG as
high as 4.37 µg m$^{-3}$. Comparing the chemical composition between different biomass burning
periods, it was apparent that biomass burning can considerably elevate the levels of organic
components, while not showing a significant effect on the production of secondary inorganic ions,
although their precursors were observed at increased levels. In addition, it's interesting that the
LG/MN ratios in different biomass burning periods were similar, while the LG/$K^+$ ratios during
the intensive BB period were abnormally higher than those in the minor and major periods. This
may be due to local soft wood combustion in the surrounding area, which have a more efficient
formation mechanism of levoglucosan than $K^+$ during flaming fires. On the other hand, this may
imply that there were other sources of $K^+$ in the study region, such as fireworks, fertilizers, soil
resuspension. Thus, based on the results from this study, governmental restrictions on biomass
burning emissions should be expanded, not only focusing on controlling open burning activities of
crop residues, but also including residential burning activities of biofuels, e.g., straws, woods or
other vegetations in the North China Plain, especially during the winter heating season.

***Data availability.*** The data used in this study are available from the corresponding author
upon request (lianglinlin@cma.gov.cn).
***Author contributions.*** LL designed conducted all observations and drafted the paper. GE
revised the paper and improved the English writing. XL drew the Fig.4 and 5. CL, WX, YC, ZD,
GZ, JS and XZ interpreted the data and discussed the results. All authors approved the final
version for publication.



*Competing interests.* The authors declare that they have no conflict of interest.

*Special issue statement.* This article is part of the special issue "In-depth study of air pollution sources and processes within Beijing and its surrounding region (APHH-Beijing) (ACP/AMT interjournal SI)". It is not associated with a conference.

*Acknowledgements.* This research is supported by the Beijing Natural Science Foundation (8192055) and CAMS Fundamental Research Funds (No. 2017Z011). The authors would like to acknowledge Yingli Yu and Ye Kuang for their help with $PM_{2.5}$ samples collection; Hongbing Cheng for help with chemical analyses.

*Financial support.* This research has been supported by the Beijing Natural Science Foundation (8192055), State Environmental Protection Key Laboratory of Sources and Control of Air Pollution Complex (SCAPC201701) and CAMS Fundamental Research Funds (No. 2017Z011). Financial support was also provided partly by the Ministry of Science and Technology (MOST) of Taiwan (MOST 103-2113-M-007-005).

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
