# Peer review of "Figure S1.* Location of Gucheng measurement station (red star) and the surrounding provinces."

_Atmospheric Chemistry and Physics, 2020_

## Referee Comment (RC1) · Anonymous Referee #1 · 24 Mar 2020

General Comments: The study of Liang et al. aimed to gain insights about the abundance of biomass burning smoke during the autumn-winter transition season, and explore the impact of biomass burning activities on the chemical properties of ambient aerosols. The data analysis was relatively straightforward, however, there were some issues with interpretation, especially there were some places with subjective interpretations. Although chemical characteristics of PM2.5 composition in minor, intensive and major biomass burning periods were presented in this study, contributions by biomass burning to PM2.5 or carbonaceous components was not quantified. There is nothing that is particularly novel in the study compared to previous work, and the environmental significance of the findings is not clear. The manuscript cannot be accepted for

publication in its current form.

Although the manuscript is not difficult to read, there are numerous grammar and language issues, which need to be addressed and improved. Several examples are listed below, Line 19: It's better to say "The measured daily average concentrations of LG, MN and K+ during this study period were 0.79±0.75 $\mu$g/m3, 0.03±0.03 $\mu$g/m3 and 1.52±0.62 $\mu$g/m3, respectively"

Line 53: "abundant" can be replaced by "extensive"

Line 84-85: "The study results demonstrate" is better to be revised to, for example, "The results of this study demonstrate", or "The results presented in this study demonstrate"

Line 133: "All measurement data quality was controlled according to standards. . .."

Line 137: "on" should be replaced by "at"

Line 197: repeated word "average" in "daily average wind speed averaged at. . .." ... . . .

Other comments:

Line30-32: Why the finding that K+ did not increased as much as LG during extensive BB episodes indicated there were other sources of K+ in the study region?

Lin 84: "GC" should be defined here.

Line 103: Why quartz filters were prebaked in such a high temperature (850°C)? This is different from the temperature widely used in other studies.

Lin 128: it should be "0.82 $\mu$g C/cm2"

Line 163: In this study, concentrations of EC were higher than SO42 , NO3- and NH4+, accounting for 13-17% of the PM2.5 mass. This is different from previous studies. Could the authors please explain why there were such high EC concentrations?

Line 170-181: There might be other reasons for the different distributions of secondary inorganic ions in different studies. For example, the major sources might be variable

with seasons and sites. Line 188-192: What about the variations of levoglucosan/OC ratio?

Line 216-220: The authors seemed to attribute the more significant differences of carbonaceous components during the nighttime vs. daytime compared to secondary inorganic ions to PBL. I can't agree. Please explain/clarify how PBL cause different accumulations or influences on carbonaceous components and secondary inorganic ions.

Line 220-223: I don't agree with the authors that there are no significant differences in chemical reactions of carbonaceous components and anhydrosugars during daytime and nighttime. For example, OC include primary and secondary organics, not only the sources but also the formation mechanisms or chemical reactions could be diffident during daytime and nighttime. Besides, levoglucosan in the atmosphere is also not stable and could undergo atmospheric chemical degradations according to previous studies.

Line 226-231: How about the difference in secondary transformations of secondary inorganic ions during the daytime and nighttime?

Line 277-279: What did the authors mean "the relationships between LG and OC, EC during daytime and nighttime were both better than those with SNA"?

Line 268-269: Are there any evidences for "frequent heating activities in form of straw burning"?

Line 287: "in the range of 1.38 to 1.82" seems only for SO2 but not for other SNA precursors.

Line 288-290: It's difficult to see the positive and negative relationships between gas precursors and PBL. Maybe better to change to scatter plots.

Line 291: If so, how to explain the lower CO concentrations in period âĔą compared to period âĔć?

Line 305-308: Seen from Fig. 4 and Fig. 5, there were also a large number of fire spots in the northwest. But why the authors stated that the air masses were "with rare biomass burning activities"?

Fig. 6: The authors presented LG together with OM. Do you mean LG is not belonging to OM?

Line 312-313: Similar to comments on Line 163, please explain why EC accounted for such a large fraction of PM2.5 mass, and stable in all days.

Line 354-359: Please revise "a time of strong process decrease in temperature". And maybe it's better to look into the reason why temperature dropped so quickly and significantly, and plot time series of temperature during these days. Besides, please add the sampling time for daytime and nighttime samples.

Line 359-361: I agree there are other sources for K+. But I'm wondering if there were fireworks and fertilizers during the study period.

Line 361-363: Since K+ is widely used as a tracer for biomass burning. It's better to illustrate from the emission characteristics of K+ and levoglucosan from biomass burning, and influences of combustion conditions (flaming and smoldering) and fuel types.

Please also note the supplement to this comment:
https://www.atmos-chem-phys-discuss.net/acp-2020-19/acp-2020-19-RC1-supplement.pdf

---

## Referee Comment (RC2) · Anonymous Referee #2 · 16 Apr 2020

In this manuscript, the authors report chemical characteristics of PM2.5 under the impact of biomass burning (BB) in the North China Plain. Several BB markers, including levoglucosan, mannosan and water-soluble potassium are measured in daytime and nighttime samples. The authors find large differences in chemical characteristics in different BB periods. They also claim that there might be other sources of potassium. My major concerns are:

1. The authors compare chemical characteristics of PM2.5 in three periods, named minor BB, intensive BB and major BB. The major BB period is from 1-23, November. As I know, the central heating system in Beijing usually starts from the middle of November.

There should be large increase in the fuel consumption, including biofuel and coal due to residential heating. I suggest to exclude the heating period from the major BB. The authors can also compare chemical compositions in the heating period with other three episodes.

2. The ratio of levoglucosan to mannosan (L/M) is wide used to distinguish different BB types. As Figure 7 showed, the L/M ratios are very stable during the whole campaign. However, the air masses are originated from different places (Figure 5) where should have different BB types. The authors should explain why the air masses passed through different BB areas have the similar L/M ratios. Moreover, recent studies have demonstrated that levoglucosan is not stable in the air and will be decomposed during atmospheric transport. How does the aging process affect the L/M ratio and its application in BB type identification?

3. For the additional sources of potassium, previous studies have demonstrated that coal combustion is the major source of potassium in Beijing, especially during winter haze. Thus, the apparent difference in levoglucosan / potassium ratios between intensive BB and other periods (Figure 7b) could be largely due to the impact of regional coal combustion.

---

## Author Comment (AC1) · 27 May 2020

**Title: "Chemical characteristics of PM$_{2.5}$: Impact of biomass burning at an agricultural site of the North China Plain during a season of transition"**

**Anonymous Referee #1**

**General Comments:**

The study of Liang et al. aimed to gain insights about the abundance of biomass burning smoke during the autumn-winter transition season, and explore the impact of biomass burning activities on the chemical properties of ambient aerosols. The data analysis was relatively straight forward, however, there were some issues with interpretation, especially there were some places with subjective interpretations. Although chemical characteristics of PM$_{2.5}$ composition in minor, intensive and major biomass burning periods were presented in this study, contributions by biomass burning to PM$_{2.5}$ or carbonaceous components was not quantified. There is nothing that is particularly novel in the study compared to previous work, and the environmental significance of the findings is not clear. The manuscript cannot be accepted for publication in its current form. Although the manuscript is not difficult to read, there are numerous grammar and language issues, which need to be addressed and improved. Several examples are listed below.

**Our reply:** We thank the referee for his/her comments. In fact, to the best of our knowledge, this study is the first one to characterize the biomass burning pollution status at a heavily polluted rural site in Hebei province during the autumn-winter transition season, following the corn harvest. The results can provide valuable source information of the biomass burning activities in the entire North China region. Moreover, we captured a unique episode with extreme biomass burning pollution, with concentrations of levoglucosan as high as 4.37 µg m$^{-3}$. Based on the multi-analysis of biomass burning molecular tracers, back trajectory analysis, fire activity data and synoptic condition, the formation process and chemical character of this severe biomass burning pollution episode were discussed in detail. In addition, combined with other chemical components analysis,

it revealed that the type of biomass burning impacts the different types of chemical components in ambient aerosol, which were have rarely been reported by previous work. Therefore, we believe that the results obtained in this study are novel and valuable for gaining additional insights into the status and impacts of biomass burning pollution in source regions of North China. Moreover, our study provides evidence to the government that it's important to pay more attention to the residential burning activities in the North China Plain, and impose burning restrictions to reduce the air pollution contributed by biomass burning.

As for the contributions of biomass burning to carbonaceous aerosol and $PM_{2.5}$, we quantified them by the molecular tracer approach and discussed the results in a companion paper, as it would render this paper too long otherwise. Nevertheless, we have added the discussion of LG/OC ratios in the revised paper.

**(1) Line 19: It's better to say "The measured daily average concentrations of LG, MN and K+ during this study period were 0.79 ± 0.75 µg m$^{-3}$, 0.03±0.03 µg m$^{-3}$ and 1.52±0.62µg m$^{-3}$, respectively"**

**Our reply:** According to the referee's comment, we changed the sentence as follows:

"The measured daily average concentrations of LG, MN and K$^+$ during this study period were 0.79 ± 0.75 µg m$^{-3}$, 0.03 ± 0.03 µg m$^{-3}$ and 1.52 ± 0.62 µg m$^{-3}$, **respectively**" (See Lines 19-20)

**(2)** Line 53: "abundant" can be replaced by "extensive"

**Our reply:** According to the referee's comment, we replaced "abundant" with "extensive" in the revised paper.

"The precursors and formation pathways of SOA from biomass burning emissions were investigated by **extensive** field observations." (See Line 51)

**(3)** Line 84-85: "The study results demonstrate" is better to be revised to, for example, "The results of this study demonstrate", or "The results presented in this study demonstrate"

**Our reply:** According to the referee's suggestion, we changed the "The study results

demonstrate" to "The results of this study demonstrate" in the revised paper.

"**The results of this study demonstrated** the biomass burning pollution status in the rural atmosphere of North China and explore the impact of biomass burning activities on the chemical properties of ambient aerosols." (See Lines 83-85)

**(4)** Line 133: "All measurement data quality was controlled according to standards..."

**Our reply:** According to the referee's suggestion, we corrected the sentence as:

"**All measurement data quality was controlled according to standard gases** (Xu et al., 2019; Lin et al., 2011; Meng et al., 2018; Ge et al., 2018)." in the revised paper. (See Lines 133-134)

**(5)** Line 137: "on" should be replaced by "at"

**Our reply:** According to the referee's suggestion, we changed the "on" to "at" in the revised paper.

"The meteorological parameters, including air temperature, relative humidity (RH) and wind speed at a 24-h resolution **at** the GC site are presented in Fig. 1" (See Lines 136-137)

**(6)** Line 197: repeated word "average" in "daily average wind speed averaged at: : :." ... : : :

**Our reply:** According to the referee's suggestion, we rewrote the sentence in the revised paper.

"while **the daily wind speed was observed with an average value of** $1.07 \pm 1.14$ m s$^{-1}$, exhibiting moist and stable synoptic conditions at this rural site." (See Lines 196-197)

**Other comments:**

**(7) Line 30-32: Why the finding that K$^{+}$ did not increased as much as LG during extensive BB episodes indicated there were other sources of K$^{+}$ in the study region?**

**Our reply:** Generally, the ratio of LG to K$^{+}$ should be consistent for the same type of biomass burning emission. Based on the published controlled biomass burning experiments, LG/K$^{+}$ ratios of straws, i.e., wheat straw, corn straw and rice straw, were measured with low values (< 1.0),

while those for soft and hard wood were averaged at high values (>10.0) (Engling et al. 2006; Cheng et al., 2013; Sullivan et al., 2008). The LG/K$^+$ ratios during periods I, III and IV (0.36, 0.52 and 0.53) in this study were similar to the observations from emission of straw burning. On the other hand, the average LG/K$^+$ ratio during the intense BB period II was observed at 1.67, significantly higher than typical straw burning ratios. This indicated that besides straw burning, there should be other types of biomass burning during the intense BB period II.

The indication of other sources of K$^+$ during the intense biomass burning period II was not suitable, and we corrected the description in the revised manuscript.

"Moreover, the LG/MN ratios in different BB periods were consistent, while the LG/K$^+$ ratio during intensive BB period was significantly elevated. This may be due to more local soft wood and smoldering combustion taking place for heating under the low prevailing temperatures; these processes tend to show more efficient formation mechanisms of levoglucosan relative to K$^+$. " (See Lines 27-31)

**(8) Lin 84: "GC" should be defined here.**

**Our reply:**  According to the referee's suggestion, we defined the GC in the revised paper.

"In this paper, we focus on quantifying multiple biomass burning tracers, i.e., LG, MN and K+ as well as other chemical species in PM$_{2.5}$ in **(Gucheng, GC)** during the autumn-winter transition biomass burning season." (See Line 82)

**(9) Line 103: Why quartz filters were prebaked in such a high temperature (850℃)? This is different from the temperature widely used in other studies.**

**Our reply:**  In this study, quartz fiber filters were prebaked at 850 °C to remove carbonaceous material. When analyzed the IMPROVE-A temperature protocol, the samples are heated as high at to 840 °C to detect OC and EC. In order to be consistent with the OC/EC analysis by IMPROVE-A temperature protocol, the quartz filters were prebaked at 850 °C to make sure all carbonaceous material was removed prior to sample collection.

**(10) Line 128: it should be "0.82 gC/cm$^2$"**

**Our reply:** According to the referee's suggestion, the detection limit of OC was corrected as **0.82 μgC cm$^{-2}$** in the revised paper. (See Line 128)

**(11) Line 163: In this study, concentrations of EC were higher than SO$_4^{2-}$, NO$_3^-$ and NH$_4^+$, accounting for 13 - 17% of the PM$_{2.5}$ mass. This is different from previous studies. Could the authors please explain why there were such high EC concentrations?**

**Our reply:** The average concentrations of SO$_4^{2-}$, NO$_3^-$ and NH$_4^+$ in this study were similar as those observed by Chi et al., (2018) at GC during winter time in 2016. Unfortunately, there is no observation of OC/EC at Gucheng during the autumn or winter season published to compare our results to. There maybe three reasons for the high EC values measured in this study. Firstly, GC is a rural site in the heavily polluted Hebei province, with the worst air quality in China, where abundant industries are located, such as power, steel, chemical industry, etc. Heavily polluted cities and regions are located throughout Hebei province, i.e., Shijiazhuang, Xingtai, Handan, Cangzhou, Hengshui, all of which are located south and southwest of the GC site. Based on the source identification of trace elements, industry is the second source in PM$_{2.5}$ at the GC site, and it contributed ~ 23% to PM$_{2.5}$ (Liu et al., 2020). Moreover, the GC station is also surrounded by agricultural fields. The observation time in this study was following the autumn crop harvest season, and agricultural diesel vehicles were widely used during this time, which emit abundant EC. In addition, when entering into November, coal combustion for heating in the North China started to increase. Thus, due to these abundant sources, i.e., industry, diesel vehicles, coal combustion, together with biomass burning, EC was observed at such high values at the GC site.

**(12) Line 170-181: There might be other reasons for the different distributions of secondary inorganic ions in different studies. For example, the major sources might be variable with seasons and sites.**

**Our reply:** We agree with this comment, and realize that there were many reasons, i.e., meteorological conditions, emissions of precursors, regional transport, etc., for the different distributions of secondary inorganic ions in different studies. We compared the values of SNA in our study with the results from Chi et al., (2018), which were also observed at the GC site, yet

during winter of 2016,; both studies found $NO_3^-$ concentrations to exceed those of $SO_4^{2-}$. However, these authors observed $NH_4^+$ to be the dominant component of SNA, while $NO_3^-$ was the most abundant ion observed in this study.

**(13) Line 188-192: What about the variations of levoglucosan/OC ratio?**

**Our reply:**   According to the referee's suggestion, we added the LG/OC ratios in Table 1, and also added the discussion of the variation of LG/OC ratios in the revised paper.

"Accordingly, the LG/OC ratio increased to 0.045 during period II, which is higher than most of the published field observations, i.e., at urban sites (Zhang et al., 2008; Cheng et al., 2013; Zhang et al., 2014), rural sites (Sang et al., 2013; Ho et al., 2014; Pietrogrande et al., 2015; Mkoma et al., 2013) and agricultural sites (Ho et al., 2014; Jung et al., 2014), yet lower than at an urban site in northern Italy during winter time (Pietrogrande et al., 2015). During the major BB (period III) and heating season (period IV), due to the combustion of coal and biofuel for heating, the organic carbon component remained at a high level (55.2 ±17.1 µg m$^{-3}$ and 69.4 ± 24.6 µg m$^{-3}$, respectively). These levels are more than 3 times of that during the minor BB period I. Due to the abundance of organic aerosols, the LG/OC ratios during periods III and IV decreased to 0.016 ± 0.005 and 0.014 ± 0.006, respectively, even lower than those in the minor BB period I (0.025 ± 0.008). " (See Lines 276-286)

**(14) Line 216-220: The authors seemed to attribute the more significant differences of carbonaceous components during the nighttime vs. daytime compared to secondary inorganic ions to PBL. I can't agree. Please explain/clarify how PBL cause different accumulations or influences on carbonaceous components and secondary inorganic ions.**

**Our reply:** In fact, we don't mean PBL can cause different influences on carbonaceous components and secondary inorganic ions. We meant carbonaceous components, except secondary organic components, as well as anhydrosugars, were mainly influenced by the variations of the PBL height. SNA can not only be impacted by PBL variations, but is also affected by the photochemical formation processes during daytime.

We discuss this in the revised paper as follows: "That may be due to the variations of

pollutant concentrations not only being controlled by the chemical reactions but also being subject to the influence of the planetary boundary layer (PBL) development. In the night, the PBL height decreases, compressing air pollutants into a shallow layer, and subsequently resulting in faster accumulation and higher concentrations of pollutants (Zheng et al., 2015; Zhong et al., 2018; 2019)." (See Lines 216-220)

**(15) Line 220-223: I don't agree with the authors that there are no significant differences in chemical reactions of carbonaceous components and anhydrosugars during daytime and nighttime. For example, OC include primary and secondary organics, not only the sources but also the formation mechanisms or chemical reactions could be diffident during daytime and nighttime. Besides, levoglucosan in the atmosphere is also not stable and could undergo atmospheric chemical degradations according to previous studies.**

**Our reply:** We agree that the chemical reactions of carbonaceous components and anhydrosugars during daytime and nighttime are not the same. In order to make the description more rigorous, we rewrote this section in the revised paper.

"Elemental carbon and primary organic components are not subject to significant differences in chemical reactions in ambient air between daytime and nighttime, and they will be mainly influenced by the variations of the PBL height. The contributions of OM and EC to $PM_{2.5}$ were observed to be higher at nighttime (53.9% and 16.6%) than daytime (43.8% and 13.7%) as well. Moreover, the chemical degradation of levoglucosan may occur due to photochemical reaction in the ambient aerosols, further lowering the levoglucosan levels in daytime (Sang et al., 2016; Gensch et al., 2018). Correspondingly, the contribution of LG to $PM_{2.5}$ during nighttime (0.64%) was observed to be higher than that during daytime (0.37%). However, secondary inorganic ions have an important formation path, i.e., photochemical processing, during daytime. Thus, the secondary species ($SO_4^{2-}$, $NO_3^-$ and $NH_4^+$) were enhanced during daytime due to photochemical formation (Fig. 2 and Fig. 3)." (See Lines 220-231)

**(16) Line 226-231: How about the difference in secondary transformations of secondary inorganic ions during the daytime and nighttime?**

**Our reply:**   The topic of this paper is about the impact of biomass burning activities on the chemical properties of ambient aerosols. The difference of secondary inorganic ion processes during the daytime and nighttime is not associated with the main focus of this paper. It is a very complex research topic and should be discussed in more detail in a separate paper. In any case, we thank the referee for this valuable comment.

**(17) Line 277-279: What did the authors mean "the relationships between LG and OC, EC during daytime and nighttime were both better than those with SNA"?**

**Our reply:**   We meant to say that the relationships between LG and OC (and EC) during daytime and nighttime, were better than the relationships between LG and SNA during daytime and nighttime. In order to make the description more clear, we have rewritten this sentence in the revised manuscript.

"Moreover, the relationships between LG and OC (and EC) were better than those between LG and SNA during daytime and nighttime *(Fig. S3).*" (See Lines 290-292)

[Figure]

***Figure S3***. Correlations between LG and OC as well as EC during (a) daytime and (b) nighttime, and scatter plot

of LG versus SNA (i.e., $SO_4^{2-}$, $NO_3^-$ and $NH_4^+$) during (c) daytime and (d) nighttime.

**(18) Line 268-269: Are there any evidences for "frequent heating activities in form of straw burning"?**

**Our reply:** We thank the anonymous referees for this comment. In fact, because the ambient levels of $K^+$ and $Cl^-$ (which are common tracers for biomass burning sources, and emissions from straw burning in particular) strongly increased during period II, , it can be deduced that heating activities occurred in form of straw burning in period II. In order to make the description more reasonable, the sentence was rewritten in the revised manuscript.

"$K^+$ and $Cl^-$, the common biomass burning tracers utilized in many studies (Duan et al., 2004; Cheng et al., 2013), were also observed with increased abundance during period II. Thus, it can be inferred that the episode on 31 October, 2016 with high $PM_{2.5}$ levels was apparently caused by intensive biomass combustion activities in the North China Plain." (See Lines 263-266)

**(19) Line 287: "in the range of 1.38 to 1.82" seems only for $SO_2$ but not for other SNA precursors.**

**Our reply:** According to the referee's suggestion, the sentence was rewritten in the revised paper.

"The precursor gases of SNA, i.e., $SO_2$, $NO$, $NO_2$ and $NH_3$, were observed to have an increasing trend when biomass burning was prevalent during periods III and IV, with the ratios to period I arranged from 1.13 to 1.90 (Table 2)." (See Lines 296-299)

**(20) Line 288-290: It's difficult to see the positive and negative relationships between gas precursors and PBL. Maybe better to change to scatter plots.**

**Our reply:** According to the referee's suggestion, we added the scatter plots figure of the relationships between gas precursors and PBL in the revised paper. **(See Figure S5)**

[Figure]

*Figure S5*. Relationships between daily average PBL and gases at GC site during the observation period.

**(21) Line 291: If so, how to explain the lower CO concentrations in period II, compared to period III?**

Our reply:   CO can be emitted from varied combustion sources, i.e., fossil fuel combustion and biomass burning. The ambient CO levels are not only impacted by different kinds of burning materials, but also influenced by meteorological conditions and regional transport. In order to make the description more rigorous, we rewrote the sentences in the revised paper.

"Combustion from different fossil fuels (coal, gasoline, diesel, etc.) and biomasses (straws, woods, leaves, etc.) can all emit CO into the atmosphere (Streets et al., 2003; Chantara et al., 2019; Merico et al., 2020). Due to the more abundant combustion in the colder weather, the concentrations of CO also increased to 1.65 ± 0.53 ppm and 1.18 ± 0.83 ppm during the major biomass burning period III and the heating season period IV, respectively." (See Lines 302-306)

**(22) Line 305-308: Seen from Fig. 4 and Fig. 5, there were also a large number of fire spots in the northwest. But why the authors stated that the air masses were "with rare biomass burning activities"?**

Our reply:   Fig. 4 reflects the total fire spots in two periods, i.e., before November (period I) and after November (period III and IV). Fig. 5 shows the back trajectories for three days (31 Oct, 1-2

Nov, 2016). In fact, we meant to say that there were rare biomass burning activities **northwest of Monglia, NOT northwest of the GC site.**

"the air masses arriving at GC were advected from the northwest of Mongolia, where mostly desert areas are present, with less farm land and rare biomass burning activities (Fig. 5)." (See Lines 320-322)

**(23) Fig. 6: The authors presented LG together with OM. Do you mean LG is not belonging to OM?**

**Our reply:** We thank the anonymous referees for this valuable comment. LG certainly belongs to OM. In order to make the description more accurate, we deleted the LG data and re-draw the Figure 6 in the revised paper. (See Lines 323-325)

[Figure]

Fig. 6. Mean percentiles of major components in $PM_{2.5}$ with respect to different biomass burning pollution periods at GC site during the sampling time.

**(24) Line 312-313: Similar to comments on Line 163, please explain why EC accounted for such a large fraction of $PM_{2.5}$ mass, and stable in all days.**

**Our reply:** With the variation of biomass burning pollution periods, the EC fraction seems to exhibit no obvious change during periods I, II and III, but slightly increased in the heating season (period IV). However, the daily mass concentrations of EC obvious variability, in the range of 2.46 - 74.9 μg m$^{-3}$. Compared to the minor period I, the average EC concentrations during other

polluted periods were all observed to be elevated (Table 2).

As for why EC accounted for such a large fraction of $PM_{2.5}$ mass in this study, there may be two main reasons. First, there were abundant sources of EC at the GC site during the observation period, i.e., industry, vehicle exhaust, coal combustion, together with biomass burning, as we replied to the earlier comment (11). Secondly, the mass concentration of $PM_{2.5}$ was reconstituted by the sum of carbonaceous components ($1.6 \times OC + EC$) and inorganic ions ($SO_4^{2-} + NH_4^+ + NO_3^- + Cl^- + Ca^{2+} + Na^+ + K^+ + Mg^{2+}$) in this study. Although this approach was widely used by many researchers, it may underestimated the actual value of $PM_{2.5}$, since there may exist some undetectable chemical components, besides carbonaceous species and water soluble ions. Thus, EC taking such large faction of $PM_{2.5}$ mass in this study appears to be reasonable.

**(25) Line 354-359: Please revise "a time of strong process decrease in temperature". And maybe it's better to look into the reason why temperature dropped so quickly and significantly, and plot time series of temperature during these days. Besides, please add the sampling time for daytime and nighttime samples.**

**Our reply:** According to the referee's suggestion, we rewrote the sentence as "There was a significant drop in temperatures at the GC site during period II, with the average daily temperature sharply decreasing from 7.5 ℃ on 30 Oct to 0.31 ℃ on 31 Oct, 20106, while the average temperature at night of 31 Oct decreased to -3.4 ℃ (Fig.1)." (See Lines 390-392)

Moreover, we added the sampling time for daytime and nighttime samples section 2.1, site descriptions and sampling.

"The daytime samples were collected from 07:00 to 19:00, while nighttime samples were collected from 19:00 to 07:00 local time of the next day." (See Lines 100-101)

**(26) Line 359-361: I agree there are other sources for $K^+$. But I'm wondering if there were fireworks and fertilizers during the study period.**

**Our reply:** We thank the anonymous referees for this valuable comment. As far as we know, there were no fireworks or obvious use of fertilizers during the observation period. We admit that tThe description of fireworks and fertilizers as other sources of $K^+$ was not suitable here, and we

corrected the discussion in the revised manuscript.

"Based on the results from previous biomass source combustion studies (Engling et al., 2006; Chantara et al., 2019), compared to $K^+$, it appears that there is a large enrichment of LG during wood burning with efficient formation of LG during the flaming phase. This high LG/$K^+$ ratio in period II is probably the most representative of the local BB aerosol, similar to the smoke aerosols from soft wood combustion (China fir and red pine) (Sang et al., 2013; 2020)." (See Lines 385-389)

**(27) Line 361-363: Since $K^+$ is widely used as a tracer for biomass burning. It's better to illustrate from the emission characteristics of $K^+$ and levoglucosan from biomass burning, and influences of combustion conditions (flaming and smoldering) and fuel types.**

Our reply:  We thank the anonymous referees for this valuable comment. We extended the discussion of the reasons for high values of the LG/$K^+$ ratio in period II, from the emission characteristics of LG and $K^+$ of different types of biomass burning and combustion conditions (flaming and smoldering) in the revised paper.

"Based on the results from previous biomass source combustion studies (Engling et al., 2006; Chantara et al., 2019), compared to $K^+$, it appears that there is a large enrichment of LG in wood burning with efficient formation of LG during the flaming phase. This high LG/$K^+$ ratio in period II is probably the most representative of the local BB aerosol, similar to the smoke aerosols from wood combustion (China fir and red pine) (Sang et al., 2013; 2020). There was a significant drop in temperatures at the GC site during period II, with the average daily temperature sharply decreasing from 7.5 °C on 30 Oct to 0.31 °C on 31 Oct, 20106, while the average temperature at night of 31 Oct decreased to -3.4 °C (Fig.1). Thus, combustion activities were apparently intense around the sampling site for heating purposes, in form of burning of straws, as well as local soft woods, since these fuels are also commonly used in rural areas of North China, i.e., pine, poplar, China fir, etc.

Moreover, LG/$K^+$ ratios also can be influenced by combustion conditions, i.e., smoldering and flaming burns. Compared to flaming combustion, smoldering burns are characterized by more efficient levoglucosan formation, whereas $K^+$ emissions are relative lower. Consequently,

relatively high LG/K$^+$ ratios were observed in smoldering combustion experiments (e.g., Schkolnik et al., 2005; Lee et al., 2010). Thus, the elevated LG/K$^+$ ratios during period II can be partially explained by more abundant smoldering combustion under the low prevailing temperatures. This is consistent with the common heating custom in the rural areas in North China, where biofuels are typically subject to smoldering combustion in residential stoves for heating purposes." (See Lines 385-402)

**References:**

Chantara, S., Thepnuan, D., Wiriya, W., Prawan, S., and Tsai, Y.I.: Emissions of pollutant gases, fine particulate matters and their significant tracers from biomass burning in an open-system combustion chamber, Chemos. 224, 407-416, https://doi.org/10.1016/j.chemosphere.2019.02.153, 2019.

Cheng, Y., Engling, G., He, K.B., Duan, F.K., Ma, Y.L., Du, Z.Y., Liu, J.M., Zheng, M., and Weber, R.J.: Biomass burning contribution to Beijing aerosol, Atmos. Chem. Phys., 13, 7765-7781, https://doi.org/10.5194/acp-13-7765-2013, 2013.

Duan, F., Liu, X., Yu, T., and Cachier, H.: Identification and estimate of biomass burning contribution to the urban aerosol organic carbon concentrations in Beijing, Atmos. Environ., 38, 1275-1282, https://doi.org/10.1016/j.atmosenv.2003.11.037, 2004.

Engling, G., Carrico, C.M., Kreidenweis, S.M., Collett Jr, J.L., Day, D.E., Malm, W.C., Lincoln, L., Hao, W.M., Iinuma, Y., and Herrmann, H.: Determination of levoglucosan in biomass combustion aerosol by high-performance anion-exchange chromatography with pulsed amperometric detection, Atmos. Environ., 40, S299-S311, https://doi.org/10.1016/j.atmosenv.2005.12.069, 2006.

Engling, G., Lee, J.J., Tsai, Y.W., Lung, S.C.C., Chou, C. C.K., and Chan, C.Y.: Size resolved anhydrosugar composition in smoke aerosol from controlled field burning of rice straw, Aerosol Sci. Tech., 43, 662-672, https://doi.org/10.1080/02786820902825113, 2009.

Gensch, I., Sang-Arlt, X.,F., Laumer, W., Chan, C.,Y., Engling, G., Rudolph, J., and Kiendler-Scharr, A.: Using δ13C of levoglucosan as a chemical clock, Environ. Sci. Technol. https://pubs.acs.org/doi/10.1021/acs.est.8b03054, 2018.

Ho, K.F., Engling, G., Sai Hang Ho, S., Huang, R., Lai, S., Cao, J., and Lee, S.C.: Seasonal variations of anhydrosugars in PM$_{2.5}$ in the Pearl River Delta Region, China, Tellus B, 66, 22577, https://doi.org/10.3402/tellusb.v66.22577, 2014.

Klejnowski, K., Janoszka, K., and Czaplicka, M.: Characterization and seasonal variations of organic and elemental carbon and levoglucosan in PM$_{10}$ in Krynica Zdroj, Poland, Atmos., 8(10), 190. https://doi.org/10.3390/atmos8100190, 2017.

Lee, T., Sullivan, A.P., Mack, L., Jimenez, J.L., Kreidenweis, S.M., Onasch, T.B., Worsnop, D.R., Malm, W., Wold, C.E., Hao, W.M., and Collett Jr., J.L.: Chemical smoke marker emissions during flaming and smoldering phases of laboratory open burning of wildland fuels, Aerosol Science and Technology 44. http://dx.doi.org/10.1080/02786826.2010.499884, 2010.

Liu, L., Liu, Y., Wen, W., Liang, L., Ma, X., Jiao, J., and Guo, K.: Source Identification of Trace Elements in PM2.5 at a Rural Site in the North China Plain, Atmos., 11(2), 179, https://doi.org/10.3390/atmos11020179, 2020.

Merico, E., Grasso, F.M., Cesari, D., Decesari, S., Belosi, F., Manarini, F., De Nuntiis, P., Rinaldi, M., Gambaro, A., Morabito, E., and Contini, D.: Characterisation of atmospheric pollution near an industrial site with a biogas production and combustion plant in southern Italy, https://doi.org/10.1016/j.scitotenv.2020.137220, Sci. Tot. Environ., 717, 137220, 2020.

Mkoma, S. L., Kawamura, K., and Fu, P. Q.: Contributions of biomass/biofuel burning to organic aerosols and particulate matter in Tanzania, East Africa, based on analyses of ionic species, organic and elemental carbon, levoglucosan and mannosan, Atmos. Chem. Phys., 13, 10325–10338, https://doi.org/10.5194/acp-13-10325-2013, 2013.

Pietrogrande, M.C., Bacco, D., Ferrari, S., Kaipainen, J., Ricciardelli, I., Riekkola, M.-L., Trentini, A., and Visentin, M.: Characterization of atmospheric aerosols in the Po valley during the supersito campaigns — Part 3: Contribution of wood combustion to wintertime atmospheric aerosols in Emilia Romagna region (Northern Italy), Atmos. Environ., 122, 291-305, https://doi.org/10.1016/j.atmosenv.2015.09.059, 2015

Sang, X.F., Chan, C.Y., Engling, G., Chan, L.Y., Wang, X.M., Zhang, Y.N., Shi, S., Zhang, Z.S., Zhang, T., and Hu, M.: Levoglucosan enhancement in ambient aerosol during springtime transport events of biomass burning smoke to Southeast China, Tellus B 63, 129-139, https://doi.org/10.1111/j.1600-0889.2010.00515.x, 2011.   .

Sang, X.F., Gensch, I., Kammer, B., Khan, A., Kleist, E., Laumer, W., Schlag, P., Schmitt, S.H., Wildt, J., Zhao, R., Mungall, E.L., Abbatt, J.P.D., and Kiendler-Scharr, A.: Chemical stability of levoglucosan: An isotopic perspective, J. Geophys. Res., 43, 5419-5424, https://doi.org/10.1002/2016GL069179, 2016.

Sang, X.F., Zhang, Z.S., Chan, C.Y., and Engling, G.: Source categories and contribution of biomass smoke to organic aerosol over the southeastern Tibetan Plateau, Atmos. Environ., 78, 113-123, https://doi.org/10.1016/j.atmosenv.2012.12.012, 2013.

Schkolnik, G., Falkovich, A.H., Rudich, Y., Maenhaut, W., and Artaxo, P.: New analytical method for the determination of levoglucosan, polyhydroxy compounds, and 2-methylerythritol and its application to smoke and rainwater samples, Environ. Sci. Technol., 39, 2744-2752, https://doi.org/10.1021/es048363c, 2005.

Streets, D.G., Bond, T.C., Carmichael, G.R., Fernandes, S.D., Fu, Q., He, D., Klimont, Z., Nelson, S.M., Tsai, N.Y., Wang, M.Q., Woo, J.H., and Yarber, K.F.: An inventory of gaseous and primary aerosol emissions in Asia in the year 2000, J. Geophys. Res., 108 (D21), 8809. http://dx.doi.org/10.1029/2002JD003093, 2003.

Sullivan, A. P., Holden, A. S., Patterson, L. A., McMeeking, G.R., Kreidenweis, S. M., Malm, W. C., Hao, W. M., Wold, C.E., and Collett Jr., J. L.: A method for smoke marker measurements and its potential application for determining the contribution of biomass burning from wildfires and prescribed fires to ambient PM2.5 organic carbon, J. Geophys. Res., 113, D22302, https://doi.org/10.1029/2008JD010216, 2008.

Zhang, T., Cao, J.J., Chow, J.C., Shen, Z.X., Ho, K.F., Ho, S.S.H., Liu, S.X., Han, Y.M., Watson, J.G., Wang, G.H., and Huang, R.J., 2014. Characterization and seasonal variations of levoglucosan in fine particulate matter in Xi'an, China, J. Air Waste Manage., 64, 1317-1327, https://doi.org/10.1080/10962247.2014.944959, 2014.

Zhang, T., Claeys, M., Cachier, H., Dong, S.P., Wang, W., Maenhaut, W., and Liu, X.D.: Identification and estimation of the biomass burning contribution to Beijing aerosol using levoglucosan as a molecular marker, Atmos. Environ., 42, 7013-7021, https://doi.org/10.1016/j.atmosenv.2008.04.050, 2008.

Zhang, Z., Gao, J., Engling, G., Tao, J., Chai, F., Zhang, L., Zhang, R., Sang, X., Chan, C. Y., Lin, Z., and Cao, J.: Characteristics and applications of size-segregated biomass burning tracers in China's Pearl River Delta region, Atmos. Environ., 102, 290-301, https://doi.org/10.1016/j.atmosenv.2014.12.009, 2015.

---

## Author Comment (AC2) · 27 May 2020

**Title: "Chemical characteristics of PM$_{2.5}$: Impact of biomass burning at an agricultural site of the North China Plain during a season of transition"**

**Anonymous Referee #2**

In this manuscript, the authors report chemical characteristics of PM2.5 under the impact of biomass burning (BB) in the North China Plain. Several BB markers, including levoglucosan, mannosan and water-soluble potassium are measured in daytime and nighttime samples. The authors find large differences in chemical characteristics in different BB periods. They also claim that there might be other sources of potassium.

My major concerns are:

**(1) The authors compare chemical characteristics of PM$_{2.5}$ in three periods, named minor BB, intensive BB and major BB. The major BB period is from 1-23, November. As I know, the central heating system in Beijing usually starts from the middle of November. There should be large increase in the fuel consumption, including biofuel and coal due to residential heating. I suggest to exclude the heating period from the major BB. The authors can also compare chemical compositions in the heating period with other three episodes.**

**Our reply:** According to the referee's suggestion, we separated the observation periods into four sub-periods, as follows: 15-30 October (Period I: Minor biomass burning), 31 October (Period II: Intensive biomass burning), 1-14 November (Period III: Major biomass burning), and 15-23 November (Period IV: Heating season).

Accordingly, Table 2, Figure 6, Figure 7 and the discussion in **section 3.3 and section 3.4** were also updated in the revised paper. **(See Lines 255-402).**

Table 2. Concentrations of chemical components in PM$_{2.5}$ aerosols and gaseous species collected at the GC site during the four biomass burning periods from 15 Oct to 23 Nov 2016.

| Species | Period I (15-30 Oct) Minor BB | Period II (31 Oct) Intensive BB | | Period III (1 -14, Nov) Major BB | | Period IV (15 -23, Nov) Heating period | |
|---|---|---|---|---|---|---|---|
| | Concentration | Concentration | Ratio* | Concentration | Ratio* | Concentration | Ratio* |
| PM$_{2.5}$ | 81.0 ± 44.5 | 235 | 2.91 | 163 ± 46.7 | 2.01 | 189 ± 83.0 | 2.33 |
| Levoglucosan | 0.36 ± 0.14 | 4.37 | 12.1 | 0.90 ± 0.37 | 2.50 | 0.96 ± 0.63 | 2.67 |
| Mannosan | 0.015 ± 0.005 | 0.18 | 12.0 | 0.038 ± 0.015 | 2.53 | 0.050 ± 0.026 | 3.33 |
| OC | 16.2 ± 7.52 | 96.3 | 5.93 | 55.2 ±17.1 | 3.41 | 69.4 ± 24.6 | 4.28 |
| EC | 12.2 ± 5.85 | 36.0 | 2.96 | 25.5 ± 10.1 | 2.09 | 36.4 ± 21.5 | 2.98 |
| TC | 28.4 ± 13.1 | 132 | 4.66 | 80.9 ± 34.6 | 2.85 | 106 ± 55.3 | 3.73 |
| SO$_4^{2-}$ | 10.3 ± 8.96 | 4.56 | 0.44 | 11.8 ± 6.02 | 1.15 | 9.08 ± 3.87 | 0.88 |
| NO$_3^-$ | 16.6 ± 12.9 | 18.1 | 1.09 | 16.5 ± 6.42 | 0.99 | 12.6 ± 5.76 | 0.76 |
| NH$_4^+$ | 10.1 ± 7.40 | 10.0 | 0.99 | 12.0 ± 4.35 | 1.19 | 10.3 ± 3.62 | 1.02 |
| K$^+$ | 1.16 ± 0.36 | 2.61 | 2.25 | 1.76 ± 0.46 | 1.52 | 1.65 ± 0.84 | 1.42 |
| Cl$^-$ | 3.46 ± 1.97 | 7.49 | 2.16 | 5.58 ± 2.16 | 1.61 | 6.27 ± 2.58 | 1.81 |
| OC/EC | 1.53 ± 0.35 | 2.67 | 1.75 | 2.31 ± 0.59 | 1.51 | 2.04 ± 0.31 | 1.33 |
| NO$_3^-$/ SO$_4^{2-}$ | 1.74 ± 0.60 | 3.96 | 2.28 | 1.50 ± 0.35 | 0.86 | 1.42 ± 0.47 | 0.82 |
| LG/OC | 0.025 ± 0.008 | 0.045 | 1.80 | 0.016 ± 0.005 | 0.64 | 0.014 ± 0.006 | 0.56 |
| LG/EC | 0.039 ± 0.019 | 0.121 | 3.10 | 0.038 ± 0.017 | 0.97 | 0.028 ± 0.013 | 0.72 |
| LG/MN | 24.9 ± 4.44 | 24.1 | 0.97 | 24.8 ± 6.46 | 1.00 | 18.3 ± 4.27 | 0.73 |
| LG/K$^+$ | 0.36 ± 0.081 | 1.67 | 4.64 | 0.51 ± 0.16 | 1.42 | 0.53 ± 0.15 | 1.47 |
| NO (ppb) | 21.7 ± 12.5 | 21.7 | 1.00 | 39.6 ± 15.4 | 1.82 | 39.3 ± 23.6 | 1.81 |
| NO$_2$ (ppb) | 21.8 ± 4.95 | 26.5 | 1.22 | 32.7 ± 7.27 | 1.50 | 24.6 ± 10.2 | 1.13 |
| NO$_X$ (ppb) | 43.6 ± 16.3 | 48.2 | 1.11 | 72.4 ± 17.8 | 1.66 | 64.0 ± 33.4 | 1.47 |
| SO$_2$ (ppb) | 5.83 ± 2.46 | 8.04 | 1.38 | 11.1 ± 4.10 | 1.90 | 9.75 ± 3.31 | 1.67 |
| CO (ppm) | 0.44 ± 0.33 | 0.70 | 1.59 | 1.65 ± 0.53 | 3.75 | 1.18 ± 0.83 | 2.68 |
| O$_3$ (ppb) | 9.79 ± 4.88 | 23.2 | 2.37 | 7.51 ± 3.87 | 0.77 | 9.59 ± 7.55 | 0.98 |
| NH$_3$ (ppb) | 14.3 ± 6.12 | 11.1 | 0.78 | 18.6 ± 8.03 | 1.30 | 21.2 ± 14.2 | 1.48 |

*: indicates that the ratios of the heating period, intense BB period or major biomass burning period were divided by those from the minor BB period.

[Figure]

Fig. 6. Mean percentiles of major components in $PM_{2.5}$ with respect to different biomass burning pollution periods at GC site during the sampling time.

[Figure]

Fig. 7. Scatter plots of (a) levoglucosan versus mannosan, (b) levoglucosan versus $K^+$

**(2.1) The ratio of levoglucosan to mannosan (L/M) is wide used to distinguish different BB types. As Figure 7 showed, the L/M ratios are very stable during the whole campaign. However, the air masses are originated from different places (Figure 5) where should have different BB types. The authors should explain why the air masses passed through different BB areas have the similar L/M ratios.**

**Our reply:**  According to the referee's comment, we clustered 161 backward trajectories of the air masses arriving at GC during the whole sampling period from 15 Oct to 23 Nov, 2016, into four groups (see ***Fig. R1***). Specifically, 50% of air masses originated from North China (group 1

and group 4), while another 32% of the air masses originated from Mongolia (group 2), having passed through North China, i.e., Inner Mongolia and Hebei provinces on their route to the GC site. A similar long and high air mass, transported from West China (group 2), across Xinjiang and Inner Mongolia provinces to the GC site, was observed during the sampling period (19%).

Straws are commonly burned in China; rice straw is popular in southern China, whereas wheat straw and corn straw prevail in northern and western China. Based on source burning studies, the LG/MN ratios from burning these crop residues were similar at a relatively high values (> 10.0), and also overlapped with those from hardwood and leaf burning (*Fig. R2*) (Engling et al., 2009; Cheng et al., 2013). Thus, the LG/MN ratios alone are insufficient to separate burning emissions from crop residuals, hardwoods or leaves. Although different straw types (rice, wheat or corn straw) are prevalent in different parts of China, the LG/MN ratios of all biofuel materials in China are characterized by relatively high values. The LG/MN ratios during minor, major and intense biomass pollution periods were observed at similar high values, i.e., 24.9, 24.1 and 24.8, respectively, which were all in the range of typical straw burning values in China.

We added the discussion of the potential reason for why the ratios of LG/MN are consistently high during periods I to III in the revised paper.

"Based on source emission studies, the LG/MN ratios from crop residue burning, i.e., rice straw, wheat straw, and other straws, were similar and characterized by high values (> 10.0), yet overlapped with those from hard wood and leaf burning (Engling et al., 2009; Cheng et al., 2013). The LG/MN ratios during minor, major and intense biomass pollution periods were observed at similar high values, i.e., 24.9, 24.1 and 24.8, respectively, which were all in the range of straw burning in China." (See Lines 357-362).

[Figure]

*Figure R1.*Variations of air mass origins to GC during the observation time from 15 Oct to 23 Nov, 2016, shown

by clusters of 48 hours backward trajectories arriving at 500m aboveground level. The numbers in each panel

indicate the percentages of daily trajectories in the sampling time with such origins.

[Figure]

*Figure R2*. The raitos of LG/MN measured in source samples (Cheng et al., 2013)

**(2.2) Moreover, recent studies have demonstrated that levoglucosan is not stable in the air and will be decomposed during atmospheric transport. How does the aging process affect the L/M ratio and its application in BB type identification?**

**Our reply:**   We admit that the chemical loss of biomass burning tracers, i.e., LG and MN, may occur in the ambient aerosols due to photochemical reaction during the aging process. Several studies have investigated the loss rate of levoglucosan in ambient air (e.g., Sang et al., 2016; Gensch et al., 2018). However, it is still not known whether mannosan will undergo similar

atmospheric reactions as levoglucosan, the concentrations of which are 1-2 orders of magnitude lower than levoglucosan in ambient aerosols. Therefore, although a change in the L/M ratio with aging process may occur in the ambient aerosols, potentially impacting the source apportionment, we still can't quantify the specific change of the L/M ratio in the real ambient environment at present. In any case, we added the discussion about the potential change of the L/M ratio during the aging process in the revised paper to make the results more rigorous and reasonable.

"In addition, atmospheric degradation of levoglucosan may occur due to photochemical reaction or other aging processes. However, it is still not known whether mannosan will undergo atmospheric reactions to a similar extent as levoglucosan, the concentrations of which are 1-2 orders of magnitudes lower than levoglucosan in ambient aerosols. Therefore, although a change in the LG/MN ratio with aging processes may occur in ambient aerosols, potentially impacting the source apportionment, assessing the specific effect on the LG/MN ratios in the real ambient environment is uncertain at present." (See Lines 370-377)

**(3) For the additional sources of potassium, previous studies have demonstrated that coal combustion is the major source of potassium in Beijing, especially during winter haze. Thus, the apparent difference in levoglucosan / potassium ratios between intensive BB and other periods (Figure 7b) could be largely due to the impact of regional coal combustion.**

**Our reply:** We admit that coal combustion may impact the levoglucosan / potassium ratios in the typical heating season in winter. Yan et al. (2017) and Yu et al. (2018) demonstrated that coal combustion can also be an important source of LG and $K^+$. However, there is no information about LG/$K^+$ ratios from coal combustion reported from previous studies. We thank the referee for this valuable suggestion, and we added coal combustion as a potential impact factor for the variation of LG/MN ratios between the four periods in the revised paper.

"Moreover, it should be noted that coal combustion can also act as a potential source of LG and MN, with the relatively low LG/MN ratios (8.3) (Yan et al., 2018). Thus, the decreasing trend of LG/MN ratios in the typical heating season can partly be caused by extensive coal combustion. However, it is noteworthy that emissions from crop straws are still expected to be dominant burning materials during the heating season at the GC site, as evidenced by the low LG/$K^+$ ratios

(0.53 ± 0.15), i.e., in the typical range for straw and hard wood burning (Cheng et al., 2013)." (See

Lines 364-370)

**References:**

Cheng, Y., Engling, G., He, K.B., Duan, F.K., Ma, Y.L., Du, Z.Y., Liu, J.M., Zheng, M., and Weber, R.J.: Biomass burning contribution to Beijing aerosol, Atmos. Chem. Phys., 13, 7765-7781, https://doi.org/10.5194/acp-13-7765-2013, 2013.

Engling, G., Lee, J.J., Tsai, Y.W., Lung, S.C.C., Chou, C. C.K., and Chan, C.Y.: Size resolved anhydrosugar composition in smoke aerosol from controlled field burning of rice straw, Aerosol Sci. Tech., 43, 662-672, https://doi.org/10.1080/02786820902825113, 2009.

Gensch, I., Sang-Arlt, X.,F, Laumer, W., Chan, C.,Y., Engling, G., Rudolph, J., and Kiendler-Scharr, A.: Using $\delta$13C of levoglucosan as a chemical clock, Environ. Sci. Technol. https://pubs.acs.org/doi/10.1021/acs.est.8b03054, 2018..

Sang, X.F., Gensch, I., Kammer, B., Khan, A., Kleist, E., Laumer, W., Schlag, P., Schmitt, S.H., Wildt, J., Zhao, R., Mungall, E.L., Abbatt, J.P.D., and Kiendler-Scharr, A.: Chemical stability of levoglucosan: An isotopic perspective, J. Geophys. Res., 43, 5419-5424, https://doi.org/10.1002/2016GL069179, 2016.

Yan, C.Q., Zheng, M., Sullivan, A.P., Shen, G.F., Chen, Y.J., Wang, S.X., Zhao, B., Cai, S.Y., Desyaterik, Y., Li., X.Y., Zhou, T., Gustafsson, Ö., and Collett, J.L.: Residential coal combustion as a source of levoglucosan in China, Environ. Sci. Technol., 52(3), 1665-1674, https://doi.org/10.1021/acs.est.7b05858, 2017.

Yu, J.T., Yan, C.Q., Liu, Y., Li, X.Y., Zhou, T., and Zheng, M.: Potassium: A Tracer for Biomass Burning in Beijing? Aerosol Air Qual. Res., 18, 2447-2459, doi: 10.4209/aaqr.2017.11.0536, 2018.